# The Order Allocation Problem and the Algorithm of Network Freight Platform under the Constraint of Carbon Tax Policy

**DOI:** 10.3390/ijerph191710993

**Published:** 2022-09-02

**Authors:** Changbing Jiang, Jiaming Xu, Shufang Li, Xiang Zhang, Yao Wu

**Affiliations:** 1Modern Business Research Center, Zhejiang Gongshang University, Hangzhou 310018, China; 2School of Management and E-Business, Zhejiang Gongshang University, Hangzhou 310018, China; 3School of Accounting and Finance, Zhejiang Vocational College of Commerce, Hangzhou 310053, China

**Keywords:** carbon tax, online freight, division of labor, order distribution, division of labor in ant colonies

## Abstract

In order to solve the problems of improper order allocation and the lack of a carbon emission constraint system in the road freight transportation industry, this paper proposed an order allocation mechanism of network freight transportation with carbon tax constraints and established an order allocation optimization model with carbon tax constraints. Based on the basic characteristics of the problem, this paper redesigns the ant colony labor division expansion model, and designs a corresponding algorithm to solve the problem. By improving the update rules of the stimulus value and the threshold value, the matching difference between the order and the driver of the network freight platform is enlarged, and the matching relation-ship is dynamically adjusted, the order allocation scheme is optimized, and a more appropriate carbon tax rate range in this industry is explored. Furthermore, the problem is solved by a 0-1 integer programming algorithm, which is compared with the algorithm designed in this paper. Through multiple numerical simulation experiments, the effectiveness and feasibility of the algorithm are verified. The experimental results show that the order allocation arrangement of the online freight platform with carbon tax constraints is more economical and environmentally friendly.

## 1. Introduction

In recent years, the emission of carbon dioxide has increased sharply, which has plunged the global climate into a huge crisis, and the main driving force for this phenomenon is economic growth [1]. In order to alleviate the deterioration of the global climate, “The future of our energy: creating a low-carbon economy”, in 2003, first proposed a low-carbon economy, that is, an economic model that improves resource utilization efficiency to reduce carbon dioxide emissions. A low-carbon economy is a development model that achieves low-carbon emissions as much as possible without affecting social and economic development [2]. The core of low-carbon economy is low-carbon, and low-carbon can be divided into many sections [3]. Throughout the existing research at home and abroad, there are mainly the following aspects: energy consumption and carbon emissions [4], economic development and Carbon emissions [5], and carbon restraint systems and carbon emissions [6].

According to the statistics of my country’s logistics network, there are currently more than 15 million trucks in China, and there are as many as 30 million practitioners in the road freight industry, but more than 90% of the 30 million freight drivers are self-operated. Overall, the entire industry lacks organization. Security and system regulation leads to the high cost and low efficiency of social logistics as a whole. Wang Chao and Kim et al. [7] pointed out in their research on China’s logistics and economic development that there is a lead-lag relationship between China’s logistics infrastructure and economic development. On the whole, the approval procedures for traditional freight vehicle plans are complex and cumbersome, which will waste a lot of time and will reduce the economic benefit-to-time ratio for both the shipper and the carrier. In addition, due to the single dispatch plan, it cannot adapt to the current multi-demand and dynamic transportation plan. According to the data given by the China Council for International Cooperation on Environment and Development, my country’s transportation industry currently accounts for nearly 1/10 of carbon emissions, and nearly 90% of this is road transportation. In other words, in order to promote low-carbon freight transportation construction, it needs to be decarbonized without compromising the economics of road transport.

A low-carbon economy not only refers to a low level of carbon emissions but also an economic form whose ultimate goal is to achieve a relative balance between carbon productivity and economic development [8,9]. Therefore, from a macro perspective, low-carbon freight is actually a distributed distribution problem. Specifically, it is to reduce the carbon emissions of the freight industry as much as possible without affecting the economic benefits of the freight industry. For freight transportation, the carbon emissions generated by different freight distribution schemes are quite different. In order to reduce carbon emissions as much as possible, it is necessary to optimize and adjust the source of freight carbon emissions. Reasonable order distribution arrangements can effectively improve resource utilization, which can not only reduce operating costs but also reduce carbon emissions. Specifically, the problem studied in this paper is to find the optimal distribution matching relationship through the evaluation of the value of the order and the evaluation of the nearby freight network car-hailing after knowing the scale and demand of the order distribution. Based on the premise of not affecting the economic benefits of the platform, carbon emissions are minimized.

## 2. Literature Review

After the low-carbon economy was proposed, the United Kingdom took the lead in launching low-carbon action, and then other countries responded and launched research on the low-carbon economy. Niu Yongping [10] believes that, in an environment where the capitalist production model has gradually become mainstream, the low-carbon economy is the economic development model that human beings are forced to adopt. McEvoy et al. [11] demonstrated that a low-carbon economy can maintain a more sustainable economy based on actual data, thereby improving employment, society, and the environment. A group of economists such as Sachs [12] proposed that a large-scale economic transformation is needed at this stage to disintegrate the negative impact of the high carbon-based economy on the environment. In order to promote China’s transition to a low-carbon economy, Jiang Rui et al. [13] quantified the decoupling effects and driving factors of carbon emissions in six major domestic industries from an industry perspective. The results show that the transportation sector has the largest growth rate of carbon emissions. Carbon emission control policies maintain the healthy development of the transportation industry.

Fu Yun et al. [14] proposed three levels of low-carbon economic development models, which are low-carbon development at the macro level, energy conservation and emission reduction at the meso level, and carbon neutral technologies at the micro level. At present, the research on carbon constraints mainly focuses on three levels of carbon funds, carbon tax policies, and carbon trading mechanisms. Jia Zhijie and Lin Boqiang [15] used the recursive dynamic computable general equilibrium model to analyze the difference between carbon tax and carbon trading. The results show that a carbon tax policy has a higher emission reduction efficiency without affecting GDP benefits. Therefore, this paper mainly considers adding carbon constraints to the network freight industry from the perspective of carbon tax policy. Elkins and Baker [16] investigated relevant cases of carbon taxes and carbon emission permits, and the final conclusion pointed out that the early overall assessment of the environmental benefits of carbon taxes was positive. Chen Ziyue and Nie Puyan [17] pointed out that a carbon levy in the production process contributes to the growth of social welfare. In addition, Lu Chuanyi et al. [18] explored the impact of carbon tax policy on China’s economy and simulated the impact of carbon tax policy on the economy by establishing a dynamic recursive model. The final result verifies that carbon tax policy has little impact on GDP but can significantly reduce carbon emissions. Wang Xin et al. [19] investigated the impact of carbon costs on the GDP of various industries and found that when the tax rate reached a high tax rate of CNY 100/ton, it had an excessive impact on some industries, and when the tax rate was only CNY 10/ton, this low tax rate has almost no impact on all industries, which shows that there are differences in the need for carbon taxes to be established for different industries. Moreover, Mandell [20] and Ricke and Drouet et al. [21] both pointed out that there is a significant gap in the existing global carbon dioxide-related policies, which leads to different appropriate carbon cost standards in different regions. To this end, Zhou Yinxiang et al. [22] explored the impact of a carbon tax on China’s transportation industry and simulated the financial changes of the transportation industry when different carbon taxes were levied through a general equilibrium model. The results show that when the carbon tax rate is CNY 50. In addition, through the research on the subdivision of transportation modes, it is found that there are differences in the carbon tax rates applicable to different transportation modes.

Traditional freight only needs to deal with the needs put forward by the customer, and it can be processed on demand. However, in the era of the rapid development of information technology, high-demand orders need to be more securely and highly integrated and matched. Information and Communication Technology (ICT) can precisely meet this need, and the empirical analysis of Eurostat data by Antonio et al. [23] has verified that the development of ICT contributes to the growth of GDP and supports sustainable economic development. At present, traditional freight is transforming into network freight with ICT. Li Ye and Yu Yuewu [24] analyzed the existing freight APP with a case study. The analysis results show that orders based on freight APP can reduce the empty distance of trucks and increase the average load. This shows that the freight APP based on internet technology can effectively reduce carbon dioxide emissions in freight behavior. Zhao Yingjie et al. [25] explored the car-free carrier model and put forward the idea of establishing a car-free carrier platform by comprehensively considering the current situation of my country’s transportation and ecological environment. However, this business model involves the information of both the consignor and the carrier, which also leads to the scheduling of the online freight platform, which is a multi-party dynamic game problem. For this reason, Xiong Zhenyu and Su Youkang [26] established a game theory based on the perspective of price coordination. The neural network decision tree prediction model based on the results shows that the cost of pricing in freight can be reduced, but only considering that the cost of pricing cannot solve the current high carbon emission pain point of the freight industry. Nasiri et al. [27] studied the cross-docking vehicle routing problem by taking large-scale urban freight as an example, proposed an MLP model, and considered cost reduction from the perspective of order allocation. The results show that this way of controlling freight at the macro level is more preferable. In addition, combined with the research on energy efficiency and economic development by Pan Xin-Xin et al. [28], it can be seen that the increase in labor input will reduce energy efficiency, and reasonable order distribution means reasonable labor distribution to some extent. It can be seen that it is urgent to carry out research on freight order allocation at present. At this stage, there are single-objective optimization and multi-objective optimization in freight order allocation research, but they mainly focus on optimizing vehicle routes, profits, and costs. For example, in order to optimize the vehicle routing problem in freight transportation, Wang Suxin et al. [29] proposed a particle-ant colony hybrid algorithm for the multi-demand point vehicle scheduling problem; in order to improve the utilization of transportation resources, Chen Jing [30] addressed the problem of road freight order allocation, a centralized decision-making model built with profit maximization as the objective function. However, in these studies, most of the objective functions are based on cost, time efficiency, etc., and the cost of carbon emissions is rarely considered. In addition to these, Guajardo [31] explored the effectiveness of collaborative transportation to achieve low-carbon freight goals. It also studies the emission reduction strategies of road freight from the perspective of carbon emission allocation and compiles the emerging literature on integrating the environment into road freight.

The above literature studies low-carbon freight from different perspectives and optimizes the operation of the freight industry from different levels, but there is no research that combines order allocation and carbon emissions to consider low-carbon freight at the same time. The application scenarios of carbon constraints have been very extensive, and there are relatively mature theories and management systems, but the formulation and policies related to the domestic freight industry are still incomplete. Based on this, this paper will study the problem of order allocation based on carbon emissions, transportation costs, and the benefits obtained for factories in the supply chain, using the network freight platform to distribute goods to distributors. First, use the freight cost of transportation tasks and the environmental cost of carbon emissions to reset the response threshold of each network freight platform carrier driver; then, set each task according to the revenue brought by the order, transportation distance, transportation volume, and order urgency. The environmental stimulus value reconstructed the order distribution expansion model of the online freight platform under the constraint of carbon tax policy. In order to reduce the cost of freight transportation as much as possible, improve the utilization rate of resources, reduce environmental pollution, and fill the application gap of some strategies in this direction.

## 3. Network Freight Platform and Its Carbon Emission Measurement Method

Network freight is an emerging industry of the internet. This new freight mode is equipped with a powerful Internet of Things information system, which can integrate vehicle and freight resources more effectively. Its operation principle is shown in Figure 1. The network freight platform belongs to the information integration center and the central control center. It integrates the information flow, logistics, and capital flow of the shipper and the actual carrier; then, it makes a reasonable order distribution and dispatch; and, finally, it completes the whole freight process.

The measurement methods of carbon emissions are mainly divided into three levels, namely, the micro case method, the meso mass balance method, and the macro emission coefficient method [32]. Among them, the case method is to measure the emission source so as to obtain the carbon emission data. In other words, the case method is based on the policy data, which is relatively suitable for some small- and medium-sized enterprises. The mass balance principle mainly uses the mathematical calculation of chemistry to obtain carbon emission data, which is applicable to some chemical enterprises and is mainly used in industrial production processes. The emission coefficient method calculates the carbon emission data according to the carbon emission equation raised by IPCC. This method calculates the carbon emission by the emission coefficient of each energy substance, which can conduct macro-control on the overall carbon emissions of a specific region and is applicable to the carbon emission accounting scenarios of countries, provinces, and cities. Since the research background of this paper is based on low-carbon urban freight transportation, the macro emission coefficient method is chosen as the main carbon emission measurement method. Network freight mainly involves short-distance road transportation. The relevant emission coefficients and energy consumption of the two main vehicles are shown in Table 1. This paper mainly adopts the emission coefficient method to calculate the carbon emissions generated in the process of cargo transportation; the specific formula is as follows:(1)$$E_{i}=\mu \cdot T_{i}\cdot d_{i}$$

Among them:$E_{i}$—Total carbon emissions from truck I transportμ—Unit of carbon emissions in kg/ton/km$T_{i}$—The total volume of goods transported by truck I, in tons$d_{i}$—The total distance of freight vehicle I during transportation, in kilometers

**Table 1 ijerph-19-10993-t001:** Emission coefficients related to the road transport industry.

Vehicle Type	Type of Energy Consumption	The Energy Consumption Situation	Unit Carbon Emission
Gasoline truck	Gasoline	0.0689 L/t/km	0.1517 kg/ton/km
Diesel truck	Diesel	0.0606 L/t/km	0.1553 kg/ton/km

Note: The data come from Xie Tianrong and Wang Jing’s “Comparative Study of Carbon Emissions in Transportation Industry”.

In addition to the calculation of carbon emissions, the calculation of the carbon reduction cost is also more complicated. Liang Jin [33] pointed out that the carbon cost is affected by various factors such as the equipment cost, labor cost, patent value, marginal benefit, investment cycle, and technology dividend. At present, due to the limitations of many immature technologies, most countries choose to levy a carbon tax as a macro carbon management method. In order to verify that imposing a carbon cost can effectively reduce carbon emissions, Zhang Caiping et al. [34] used the carbon emission cost calculation method to conduct an empirical analysis in a case company, and the optimization results showed that carbon emissions were reduced by 405,021.97 tons. However, this management method is based on the measured carbon emissions. Therefore, this paper adopts a relatively extensive method to calculate the carbon cost (carbon emissions multiplied by the carbon tax rate) in order to achieve a reasonable carbon dispatch through this management and control method—the purpose of the resource.

## 4. Review of Problem Cases and Methods and Theories

### 4.1. Background

For example, in the supply chain, suppliers transport materials to distribution points. Due to the small and decentralized demand of distributors, the general transportation arrangement will expand logistics costs and produce unnecessary carbon emissions at the same time. The network freight platform can exactly solve such scattered demand. Suppliers only need to publish tasks on the freight online car-hailing platform, distribute orders through the network freight platform, and transport the goods from the supplier network to each distribution point. Considering that they are all dispersed tasks with small transport volumes, several models with smaller vehicle loads are selected. For example, the six models in Table 2 are small, medium, and large vans and small, medium, and large flatbed vehicles with load loads of 1 t,1.5 t, 2 t, 1 t, 1.8 t, and 2.2 t, respectively. The network freight platform will allocate appropriate models of vehicles according to the actual needs of distribution points.

### 4.2. Case of Network Freight Platform

Under the general background of the internet era in the 21st century, network freight arises in response to the situation. The rise of network freight benefits from the high adaptability of the mode of “Internet + in-city freight” and today’s digital technology. Internet freight transportation is a freight transportation mode that relies on the internet platform to integrate and configure transportation resources, sign a transportation service contract with the shipper as the carrier, entrust the actual carrier to complete the transportation task, and take responsibility for the actual carrier. Liu Chunsheng [35] mentioned that the three advantages of an online freight transportation platform are information visualization management, the digital sharing of transportation capacity, and industrial internet expansion, which can further promote the construction of smart logistics to a large extent. In other words, the essence of the network freight platform is a car-cargo matching platform based on internet technology, which informationizes the car-cargo resources through digital technology, integrates the information through big data processing means, and finally completes the freight handover between the shipper and the carrier. The position distribution of transportation application scenarios is roughly shown in Appendix A. Each icon in the map represents the position coordinates of the supply and demand sides in the freight process, including the position of the carrier that can accept scheduling, and the starting point and ending point of the order placed by the shipper.

In summary, the main characteristics of network freight are as follows:(1)Demand: The user demand points based on the network freight platform are decentralized, and the user’s demand is relatively large. The user’s demand position can be regarded as a collection of points.(2)Supply: The location of the carrier of the online freight platform is also highly dispersed, and there are many models of the carrier’s trucks, which can greatly meet the transportation needs of the demander.(3)Region: In the process of cargo delivery, most of the orders of the carrier are delivered within the same city, and there is little long-distance transportation. Considering the operating standards of the carrier’s drivers, not all of them are full-time freight drivers, so there are certain requirements for the transportation distance. Currently, most of them are short-distance transportation.

### 4.3. Method Theory

#### 4.3.1. Group Intelligent Division of Labor Method

The definition given by Baker [36] is a collection of individuals with interdependent relationships, where members actually depend on each other to achieve individual and collective goals. At the same time, Baker explained that this interdependence can also be understood as the meaning of division of labor. In fact, the purpose of division of labor is to better achieve common goals or to better accomplish group tasks. In the whole process, each participant plays a different role and finally maximizes the common purpose among the groups.

This order distribution problem is essentially a labor division problem, which is defined by Durkheim [37] as complex division of labor activities that have different properties and are common and mutually indispensable tasks for a group (Nunes, Mitiche) [38]. In the research field of multi-agent task assignment, the classification method is divided into two types: centralized and decentralized, and a classification method based on the time window and order constraints is extended (Dorigo et al.) [39]. With positive feedback and constructive greedy heuristics, a new computing paradigm—ant colony system—is proposed. Bonabeau et al. [40] proposed a fixed response threshold model when improving the ant colony algorithm (FRTM); this model greatly enhances the positive feedback mechanism of the ant colony algorithm. Bonabeau et al. [41] proposed an ant colony algorithm based on the stigmergic communication paradigm according to the distributed control behavior of the ant colony. Ju Chunhua and Chen Tinggui [42] introduced the extension of matter-element theory [43], which improves FRTM, and this extension makes the model more in line with the real division of labor dispatch.

The core of the labor division of swarm intelligence is the self-organization model, and the complete self-organization process is controlled by three rules, namely, state transition rules, global information update rules, and local information update rules. In other words, its improvement is basically optimized from the aspects of algorithm convergence and continuous space solution. Taking the ant colony labor division algorithm as an example, the MAX-MIN Ant System Antalgorithm [44], Continuous Ant Colony Antalgorithm [45], Genetic Algorithm-Ant algorithm [46], etc. are optimized for these two aspects. According to the above research, the current swarm intelligent labor division can be applied to problems in scenarios such as workshop scheduling [47], logistics distribution [48], and benefit distribution [49], and there are still many improvement methods in the field of swarm intelligent labor division.

#### 4.3.2. Fixed Threshold Response Model Based on Ant Colony Division of Labor

By further exploring the pheromones in the ant colony algorithm, Bonabea et al. proposed a fixed response threshold model (FRTM) for self-organizing the division of labor in ant colonies. The core point of the model is that the task is assigned with a stimulus value *s*, and the stimulus value is dynamic and will change with time, task completion progress, etc. The task with a large stimulus value is more attractive in the task set than the task with a small stimulus value. The corresponding fixed response threshold *θ* was set according to the individual characteristics of ants. The ants with a high threshold were more difficult to respond to in the ant colony than those with a low threshold. Whether ants participate in the task is determined by the stimulus value *s* of the task and the threshold value *θ* of the individual ant. When the stimulus value of a task is greater than the threshold value of the ant, the probability of ants participating in the task is higher, and vice versa. When the number and progress of ants participating in a task are lower than expected, the stimulus value of the task will increase to attract more ants to participate in the task. On the contrary, when the completion progress is higher than expected, the stimulus value of the task will increase It will gradually decrease until the stimulus value returns to 0. When the stimulus value of the task is 0, the task is completed. 

1.Environmental stimulus value changes with time

The environmental stimulus is a characteristic variable measuring the potential value of tasks. In an application scenario, if there is a task, there will be a corresponding value of the environmental stimulus, and the value of the environmental stimulus will change with time. The formula for the change in the environmental stimulus value is as follows:(2)$$s\left(t+1\right)=s\left(t\right)+\delta -\varphi \cdot n_{act}$$

In the formula, *t* represents the time variable, and *s*(*t*) is the stimulus value of the task at time *t*. In order to prevent some tasks with low stimulus value from having no ants to participate in the task, a unit time increase of stimulus value *δ* is established. In this way, if no ants participate in the task, the stimulus value will continue to increase, and the task amount will decrease during the execution of the task, so a negative variable (−*φ*·*n*) is set up; *φ* is the working efficiency of the ant, and *n_act_* represents all the ants that participated in the task.

2.Response of inactive ants to environmental stimuli

Whether ants participate in the task is affected by both the stimulus value and the threshold value. Both the stimulus value and the threshold value are compared with their own cluster. The activity participation probability formula of ants in inactive states is as follows:(3)$$P(X_{i}=0\to X_{i}=1)=\frac{{\left(S_{j}\right)}^{n}}{\left[{\left(S_{j}\right)}^{n}+{\left(\theta_{i}\right)}^{n}\right]}$$
where $X_{i}$ is used to represent the state of ant *i*, and $X_{i}=0$ represents its current idle state; $X_{i}=1$ represents that they are currently participating in the mission. $P(X_{i}=0\to X_{i}=1)$ represents the ants from an idle state to a state of participating in the probability of the task status. The probability of ants to participate in the task at this time has nothing to do with the threshold, the introduction of index *n*, and the need to control the threshold of the function curve. So, usually, type *n* = 2.

According to the response probability Equation (3), the threshold is *θ*. When the stimulus value is fixed, the higher *s* is, the greater the probability of response will be, and vice versa. When the stimulus value *s* is fixed, the threshold value is *θ_i_*. The larger the ant is, the smaller the response probability is, and vice versa, and the response probability is an intuitive indicator to measure whether the ant participates in the task.

3.The probability of an active ant quitting the task

When an individual *i* is in the state of participating in the task, it will decide whether to quit the task according to the following probability formula:(4)$$P(X_{i}=1\to X_{i}=0)=p$$

Bonabeau and others took into account the complexity of quitting tasks in actual scenarios, so *p* was generally set as a fixed constant value to simplify the model. In the fixed response threshold model, the probability of individual participation and that of quitting tasks were independent of each other.

This paper aims to promote the development of low-carbon freight transportation and takes low-carbon emissions as environmental constraints. By improving the fixed threshold response model based on the division of labor of an ant colony, it is further extended to the order allocation problem of an online freight platform in order to reduce the carbon emissions of the online freight industry.

#### 4.3.3. Applicability Analysis of Research Methods

The swarm intelligence division of labor is the overall intelligent behavior of a group composed of many simple *agents* that spontaneously cooperate to complete a common task [50]. It is an optimal or approximately optimal solution to a problem-solving method. Its applicability to the research questions of this paper is analyzed as follows:(1)Self-organization of a group task assignment

The core of the ant colony’s labor division lies in its self-organization, that is, in this labor division scenario, whether an agent participates in a task is only determined by the individual’s threshold and the task’s environmental stimulus value. The order in the online freight platform order allocation problem is equivalent to the task in the ant colony labor division problem, which will generate an environmental stimulus value to attract the online freight platform carrier driver to participate in the order, and the online freight platform carrier driver is equivalent to an individual and sets the threshold according to its own order acceptance standards and the cost of transportation operations. In this process, it is not equivalent to the traditional top-down central decision making, but it first decides whether to participate through the evaluation of the agent’s order receiving standards and transportation operation costs and then realizes the dispatch of orders according to the platform selection algorithm, which is a central-distributed decision.

(2)Environmental stimulus value and order reward

In the basic ant colony division of labor fixed response threshold model, the environmental stimulus value is the decisive information, which can highlight the value of the task. Under the same conditions, when the task value is high, that is, when the environmental stimulus value is high, the probability of attracting ants to participate in the task will be higher; otherwise, it is difficult to attract ants to participate in the task. The environmental stimulus value is a dynamic variable, which will be updated continuously with time. There is a time increment and a negative increment of the task completion progress in the update process. When the task progress is approaching saturation, the environmental stimulus value will gradually change. When the value of the task is not enough to attract the individual ants to participate in the task, the progress of the task is stagnant, and the time increment will continue to increase the environmental stimulus value, thereby attracting ants to participate in the task. In the order distribution, when the remuneration is very generous, it is easy to attract the carrier driver to take the order. On the contrary, when the order remuneration is not enough to attract the carrier driver to take the order, the platform will give some financial subsidies to the driver to ensure that the order can be successfully delivered.

(3)Individual thresholds and driver order acceptance criteria

In the basic ant colony labor division fixed response threshold, each ant has a threshold, which is used as an evaluation index to measure whether to respond to the task. Ants have different perceptions of different environmental stimuli. Ants with a low threshold have a greater probability of responding to the task, while ants with a high threshold have a relatively small probability of responding to the task. Specifically, when the online freight platform allocates orders, the carrier driver will make order-taking decisions based on metrics such as order-accepting distance and order transportation volume. These metrics are actually the key pieces of information to determine the threshold. The lower the final threshold, the higher the probability of participating in the task of cargo transportation; on the contrary, the probability of participating in the task is low, which reflects the difference in the degree of the response of different truck drivers to the task.

In summary, the division of labor in the ant colony can make the agent collectively complete all the division and cooperation from the bottom to the top without knowing the overall demand. By distinguishing the different task expectations of the agent, the agent has different tasks and then achieves a reasonable balance of the division of labor. The division of labor is actually task allocation to a certain extent, and setting the response threshold and stimulus value for the corresponding subject can improve the flexibility of task allocation and complete task allocation under the condition that the overall demand is met to the greatest extent. The order allocation problem explored in this paper is essentially the matching of vehicle and cargo resources, which finally completes the order allocation by comprehensively processing the needs and expectations between the shipper and the carrier. Although the field of swarm intelligence labor division has been applied to many scenarios, the research on using the swarm intelligence method to solve order allocation is still relatively scarce. Combined with the above applicability analysis, it can be seen that the order allocation problem has a high similarity with the ant colony labor division, so this paper chooses the threshold response model based on the ant colony labor division to analyze the order allocation problem of the online freight platform under the constraint of carbon tax optimization.

## 5. Order Allocation Modeling of an Online Freight Platform under the Constraints of Carbon Tax Policy

### 5.1. Variable Description

The symbol and decision variables are:(*n*
*agents*, *m* suborders, (*n* < m) i∈m,j∈n)Think of the position and route in question as graph *G* = (*dot, side*)*dot* = {($x_{i},y_{i}$), ($x_{0j},y_{0j}$), ($x_{1j},y_{1j}$),}. The sets of the three points are, respectively, the position of *agenti*, the starting position of the order, and the ending position of the order. side = ($l_{j},L_{ij}$). $l_{j}$ is the edge connecting the starting point of the order with the destination point, and $L_{ij}$ is the edge connecting the position point of the truck driver to the starting point of the order.The total task *T* = $\sum T_{j}$, the total task can be divided into the j subtask and the subtask $T_{j}$ = ($\varphi_{j},d_{j},G_{j},$);$\varphi_{j}$ is the degree of urgency of order j;$d_{j}$ is the distance (km) from the starting point to the destination point of order j;$G_{j}$ is the freight volume of order j (tons);$r_{j}$ is the unit remuneration (yuan/ton/km) given by order j;$v_{i}$ is the maximum capacity of *agenti* (tons);$c_{i}$ is the unit cost of *agenti* shipping (yuan/ton/km);$D_{i}$ is the distance of *agenti* to the factory in kilometers;$E_{ij}$ is the carbon emissions (kilograms) generated by *agenti* in transport task j;$e_{i}$ is the weight (in tons) of *agenti* when unloaded;$\mu_{i}$ is *agenti* ‘s carbon footprint per kilogram per ton per kilometer;*w* is the carbon tax (yuan/ton);*α* is the transportation cost transformation coefficient;*β* is the conversion coefficient of the carbon emission cost;*γ* is the emergency degree transformation coefficient;$\theta_{ij}\left(t\right)$ is the response threshold of *agenti* to task j at time *t;*$s_{j}\left(t\right)$ is the environmental stimulus value released at time *t*;δ is the self-increasing constant of the environmental stimulus per unit time;M is the increment of the control threshold.

### 5.2. Research Hypothesis

According to the problem studied in this paper, the author intends to outline the connection between the various elements in the research problem in the theoretical framework, as well as the structure, direction, and strength of the connection, as shown in Figure 2:

Based on the above research summary, this paper proposes the following hypotheses:

**H1.** 
*The carbon tax is negatively correlated with the carbon emissions of the online freight industry;*


**H2.** 
*The carbon tax is negatively correlated with the no-load distance in the online freight industry.*


**H3.** 
*The carbon tax is positively related to the carbon cost of the online trucking industry.*


**H4.** 
*The carbon tax is negatively related to the benefits of the online trucking industry.*


**H5.** 
*The no-load distance of the online freight platform is positively correlated with carbon emissions.*


**H6.** 
*The no-load distance of the online freight platform is positively correlated with the carbon cost.*


**H7.** 
*The no-load distance of the online freight platform is negatively correlated with the revenue.*


**H8.** 
*The carbon emissions of online freight platforms are positively correlated with carbon costs.*


**H9.** 
*The carbon cost of the online freight platform is negatively correlated with the benefit.*


### 5.3. Order Distribution Model of an Online Freight Platform under the Constraints of Carbon Tax Policy

In order to simplify the model, the Euclidean distance algorithm is used for the calculation of $d_{j}$, and the same is used for $D_{i}$.
(5)$$d_{j}=\sqrt{\left(x_{0j}{}^{2}-x_{1j}{}^{2})+(y_{0j}{}^{2}-y_{1j}{}^{2}\right)}$$
(6)$$D_{i}=\sqrt{\left(x_{i}{}^{2}-x_{0j}{}^{2})+(y_{i}{}^{2}-y_{0j}{}^{2}\right)}$$
(7)$$s_{j}\left(t\right)=\frac{G_{j}\cdot d_{j}\cdot r_{j}}{\gamma \cdot \varphi_{j}}$$
(8)$$S\left(t\right)=\sum s_{j}\left(t\right)$$

In the formula, $s_{j}\left(t\right)$ is the stimulus value generated by task j at time *t*, where $G_{j}\cdot d_{j}\cdot r_{j}$ is the reward paid by the customer to the agent who completes task *j*. The stimulus value is related to the number of goods transported. The transportation distance and the unit reward are proportional. $\varphi_{j}$ represents the urgency of the order (1 < $\varphi_{j}$ < 3), which is measured according to the delivery time. The more urgent the order, the smaller the value of $\varphi_{j}$, and the stimulus value is inversely proportional to the urgency of the order relation. Therefore, when the order urgency degree is similar, the stimulus value of the task with a higher total reward will be larger. Similarly, when the task urgency degree is similar, the order with a higher order urgency degree will be preferentially responded to. S(t) is the environmental stimulus value under the level of the total order. Considering that the total order is composed of multiple sub-orders, all sub-order stimulus values are taken as environmental stimulus values.
(9)$$s_{j}\left(t+1\right)=\{\begin{array}{c} s_{j}(t)+\delta ,\;\;\;\;\sum_{j}X_{ij}=0\;\\ 0,\;\;\;\;\;\sum_{j}X_{ij}=1 \end{array}$$
(10)$$S\left(t+1\right)=\sum s_{j}\left(t+1\right)$$
where δ is the fixed self-increment of the environmental stimulus per unit time, and $s_{j}\left(t+1\right)$ is the update value of the stimulus value. Considering that some tasks have stimulus values lower than all agent thresholds, an auto-increment constant of the environment per unit time is added, so the stimulus value will accumulate. In addition, considering that the task will be completed by the agent slowly, the order stimulus value will return to zero when the order is assigned. Finally, these changes are dynamic based on the initial stimulus value. When the environmental stimulus value under the total order level is zero, the entire order allocation is declared complete.
(11)$$E_{ij}=(e_{i}\cdot D_{i})\cdot \mu_{i}+d_{j}\cdot (v_{i}+2\cdot e_{i})\cdot \frac{G_{j}}{v_{i}}\cdot \mu_{i}$$

In the formula, $E_{ij}$ is the carbon emission generated by *agenti* performing task *j*, which is proportional to the transportation distance and transportation volume. The larger the values of these parameters, the higher the carbon emission and the higher the corresponding environmental cost. Among them, $e_{i}\cdot D_{i}\cdot \mu_{i}$ represents the carbon emissions generated when agenti goes to the factory with no load, and $d_{j}\cdot$($v_{i}$+$2\cdot e_{i}$)$\cdot \frac{G_{j}}{v_{i}}\cdot \mu_{i}$ represents the carbon emissions required to complete the subsequent tasks.
(12)$$C_{ij}=D_{i}\cdot e_{i}\cdot c_{i}+d_{j}\cdot (v_{i}+2\cdot e_{i})\cdot \frac{G_{j}}{v_{i}}\cdot c_{i}$$

In the formula, $C_{ij}$ is the fuel consumption cost generated by *agenti* performing task *j*, which is proportional to the transportation distance and transportation volume. $D_{i}\cdot e_{i}\cdot c_{i}$ represents the fuel consumption cost generated during the order receiving process. $d_{j}\cdot v_{i}2\cdot e_{i}\cdot \frac{G_{j}}{v_{i}}\cdot c_{i}$ represents the fuel consumption cost during the order delivery process.
(13)$$\theta_{ij}=\alpha \cdot C_{ij}+\beta \cdot E_{ij}w$$

$\theta_{ij}$ is the response threshold of *agenti* to task *j* at time t, where *α*, *β*, and *γ* are cost conversion coefficients, $C_{ij}$ represents the transportation cost generated during the entire task process, and $E_{ij}\cdot w$ represents the carbon emission environment generated during the entire task process. The cost, the threshold of *agenti* for task *j*, is jointly affected by the two.
(14)$$\theta_{ij}\left(t+1\right)=\{\begin{array}{c} \theta_{ij}(t)\;\sum_{i}X_{ij}=0\;\\ \;M\;\sum_{i}X_{ij}=1 \end{array}$$
where *M* is a maximum value, whose value is close to infinity, which means that when the *agent* has been assigned tasks, the *agent* will have no time to take care of the distribution of other orders. At this time, the threshold value of the *agent* will rise to a very high value, thus reducing the probability of the *agent* participating in other orders. On the contrary, if the agent is not assigned tasks, the threshold will remain the same.

The extended model is similar to the basic model, In the extended model, the probability of whether the *agent* participates in the task is determined by both the threshold and the environmental stimulus value.$\;X_{i}=0\to X_{i}=1$ means transition from state 0 to state 1, $X_{i}=0$ means the state of not participating in the task, and $X_{i}=1$ means the state of participating in the task.
(15)$$P(X_{i}=0\to X_{i}=1)=\frac{{\left(S_{j}\right)}^{n}}{\left[{\left(S_{j}\right)}^{n}+{\left(\theta_{ij}\right)}^{n}\right]}$$
(16)$$P(X_{i}=1\to X_{i}=0)=p$$

Since Bonabeau et al. [13] considered that the main variable is the environmental stimulus value in the actual situation, they introduced the exponent *n*, and the function image that needs to control the threshold value is a curve, therefore, the exponent *n* in the formula is usually 2. In general. At the same time, considering the complexity of truck driver exit task factors, for the simplified model, *p* is the preset constant value (0 < *p* < 1). The probabilities in Equations (15) and (16) are independent of each other, just like the basic model.

In order to verify the effectiveness of the model and the algorithm, carbon emissions, no-load distance, and total order revenue are selected as the measurement indexes with which to test the effectiveness of the model, corresponding to, respectively,$\;O_{1}$*,*
$O_{2}$, and $O_{3}$.
(17)$$O_{1}=\sum E_{ij}\cdot X_{ij}$$
(18)$$O_{2}={\left(\sum X_{ij}\cdot D_{i}\right)}^{2}$$
(19)$$O_{3}=\sum [(G_{j}\cdot d_{j}\cdot r_{j})-(C_{ij}+E_{ij}\cdot w)]\cdot X_{ij}$$

When *agenti* participates in the order, *j* does, and $X_{ij}=0;$ otherwise, $X_{ij}=1$. After all orders are completed, the distribution of orders, the carbon emissions, the no-load distance, and the total revenue of orders are counted. Combined with the actual application scenarios, since the gap between the total no-load distance in this problem is not large, its value is squared in Equation (18) to magnify the difference. Considering the end for the network freight platform, the benefits have positive value while the empty distance and carbon emissions have negative values, so they are calculated through Equations (17)–(19), and finally through the statistics Changes in the measurement indicators when different carbon tax rates are levied, and further find a more appropriate carbon tax rate range in the context of this problem.

### 5.4. Algorithm Implementation

The specific steps are as follows:

Step 1, set the following variables: the initial simulation time *t* = 0, total order demand *T*, sub-order $T_{j}$ = ($\varphi_{j},d_{j},G_{j}$), agent position and sub-order starting point and end point dot, truck weight $e_{j}$, truck maximum carrying capacity $v_{i}$, unit cost $c_{i}$, unit remuneration $r_{j}$, carbon emission coefficient $\mu_{i}$, and conversion coefficients *α*, *β*, and *γ*. The increment of self-increment *δ*, the exit probability *p*, and the regulation threshold M are determined.

Step 2, according to Equations (5)–(7), the environmental stimulus value at *t* = 0 and the task stimulus value of all sub-orders were obtained.

Step 3, calculate the response thresholds of different *agents* to different sub-orders according to Equations (5), (6) and (11).

Step 4, according to formulas (15) and (16), calculate the *agents’* state transition probability and the situation of participating or not participating in the order;

Step 5, compares the state transition probability of *agents* that can participate in the task calculated in the fourth step and arranges vehicle transportation in sequence according to the emergency degree ranking of sub-orders.

Step 6, set up a taboo table and add *agents* and orders that have received orders into the taboo table.

Step 7, update *S (t + 1)* according to Equations (8)–(10). If *S (t + 1)* > 0, update the value of $s_{j}\left(t+1\right)$ and $\theta_{ij}\left(t+1\right)$ and proceed to Step 4; otherwise, proceed to Step 8;

Step 8, calculate the measurement index according to Equations (17)–(19);

Step 9, statistics and output simulation results.

## 6. Numerical Experiment and Discussion

### 6.1. Fault Description and Parameter Settings

A large factory *X* has multiple distribution outlets for track parts in *Z* city, and the factory has corresponding supply channels for each distribution outlet. There are n distribution outlets in the city; the order task of each distribution outlet corresponds to a $T_{j}$. Now, the total transportation task (*T*) of the large factory should be completely allocated. There are *m agents* waiting to receive orders near the factory. *agenti* comes in different sizes and models (in terms of maximum deadweight, transport cost, and carbon emission factor); the conceptual diagram of the problem is in Appendix A.

The number of distribution network points in factory *X* is *n* = 18, the number of vehicles in the idle state near factory *X* is *m* = 24, and the number of six models is equal. After empirical investigation, after an example analysis of the data range in the problem, in order to ensure that the results of the problem solving are not random, the data taken in this paper are subject to different uniform distributions to more effectively verify the effectiveness of the model and algorithm, as follows: $D_{i}(i=1,\;2\ldots 12)=uniform\left(0,\;5\right),d_{j}\left(j=1,\;2\ldots 10\right)=uniform\left(10,\;100\right),G_{j}\left(j=1,\;2\ldots 10\right)=uniform\left(5,\;10\right),\varphi_{j}(j=1,\;2\ldots 10)=uniform\left(1,\;3\right)$.

Since the recent oil prices have changed greatly, the oil price in December 2021 is selected to refer to the transportation cost, and the transportation cost of gasoline and diesel cars is 0.5 and 0.4, respectively. The carbon emission coefficient $c_{i}$ is 0.151 and 0.155, and the unit remuneration $r_{j}$ is set as 4, according to the delivery pricing of different models of trucks in 2021. The self-weight of small, medium, and large vans and flat trucks is set $e_{i}$ as 0.6, 1, 1.5, 1, 1.5, and 1.8, and the maximum carrying capacity $\mu_{i}$ is 1, 1.5, 2,1, 1.8, and 2.2, respectively. Combined with the international carbon tax rate range, the carbon tax w is set as 40, the self-increment per unit time *δ* = 30, the exit probability *p* = 0.3, the transport cost conversion coefficient *α* = 1, the carbon emission conversion coefficient *β* = 20, the order urgency conversion coefficient *γ* = 5, and the increment of regulation threshold *M* = 10,000. The relevant data are shown in Table 3 and Table 4:

### 6.2. Basic Assumptions

(1)Considering the small amount of transportation but the chaotic distribution locations, it is assumed that only one agent can complete the distribution of an order, that is, there is a one-to-one correspondence between the order and agent;(2)Considering the completion efficiency and order delivery time, it is assumed that an agent will not participate in the distribution of other orders after completing an order;(3)Considering that this paper focuses on limiting carbon emissions, it is assumed that other costs not considered in this model are not calculated.

### 6.3. Result Analysis

According to the above model, the simulation experiment in this chapter is carried out on a MacBook Air laptop with an Intel(R) Core (TM)i5CPU processor and 4GB(RAM) memory. The experimental environment is the Windows 7 flagship operating system, and the algorithm is realized by MATLAB R2018 programming. The running results are as follows:

It can be seen from Table 5 that, at the 12th moment, the environmental stimulus value returns to 0, that is, the total orders of the *X* factory are completely allocated to the drivers of the network freight platform after the program cycle of 12 moments. In the whole process of order allocation, a total of 18 *agents* finally participate in order transportation. It is not difficult to see that the environmental stimulus value is inversely proportional to the quantity of remaining orders to be delivered, that is, the environmental stimulus value can well reflect the overall order task quantity at the current time *t*. In addition, when no orders have been assigned to *agents*, the environmental stimulus value will also slowly increase. Until *agents* are assigned to orders, the environmental stimulus value will slowly decrease. It can be seen that, at *t*12, the remaining orders to be delivered and the environmental stimulus value are zero. This indicates that the last 18 *agents* jointly completed the order assignment task under the model algorithm, and the whole order assignment process was completely finished at this time.

Figure 3 represents the participation of different models of cargo web carriers in order delivery, and by its variation, there is a large difference in the participation of web cargo platform carrier drivers of six models in order delivery. Since the order stimulation value at the initial moment may not be enough to stimulate the *agent* to participate in order delivery, no *agent* responds to the order delivery task at moment *t*1, while, with the update of the order stimulation value, it can be seen that, at moment *t*2, an *agent* starts to deliver the order. Since each *agent* has a different response threshold for each order, the probability of responding to the same order varies between *agents*. Because all the agents are within 5 km of factory *X*, the response threshold is largely determined by the model, and there are 6 models among 24 *agents*; each model is 4. This shows that the differentiation of the threshold value makes each order find a more suitable *agent* to transport according to its own demand in the order distribution process. Observe the order distribution participation: before the *t*4 moment, only minivans are involved in the order distribution, and 100% of minivans are involved, which indicates that, compared with other models of *agents*, the threshold of minivans is smaller. In other words, its cost is relatively low, so all minivans respond to order distribution at the *t*3 moment, and the total order distribution progress is 1/3 completed at this time. By the time of *t*6, medium and large vans as well as small flatbed trucks have not yet responded to the order distribution; at this time, the models in transportation are small vans as well as medium and large flatbed trucks, the *agents* of the three models account for 1/3 of all vehicles in transportation, and the total order distribution progress has been completed by 2/3 at this time. At the last *t*12 moment, only the small flatbed truck has not responded to the order distribution, and, finally, the small, medium, and large vans and the medium and large flatbed trucks involved in the order distribution account for 2/9, 2/9, 1/9, 2/9, and 2/9 of all vehicles involved in the order distribution, respectively, at which time the total order distribution is finished. This shows that the model will assign to the most suitable *agent* according to the different demands of the order, and at the same time, the model will finish assigning all sub-orders with the goal of completing the distribution of the total order. By repeatedly running this model, the final goal of the complete distribution of total orders is achieved, which shows that this model can effectively solve the order distribution problem in the current context and that the model is constructed with reasonableness.

Figure 4 shows the changes of some variables in the process of order allocation by the extended model algorithm program. It can be seen from Figure 4b that as time goes by, more and more agents have been allocated to the order distribution, and at the t12 when it reaches the peak value of 18, all the orders are delivered by the responding *agents*. At the same time, as shown in Figure 4c, it can be seen that the change of the overall *agent’s* busyness and idleness is consistent with the change trend of the *agents* that have participated in the order delivery. That is, at moment T12, 18 *agents* out of 24 are providing one-to-one delivery service for 18 sub-orders, and the overall *agent’s* busyness degree reaches the maximum value of 0.75. This shows that the model can effectively coordinate the distribution of orders among *agent* groups and maximize the busy degree of *agent* groups. By repeatedly running the algorithm implementation program of the model, the busyness degree between *agent* groups can be raised to a peak of 0.75, which reflects the good stability of the model.

Figure 4a shows the change of the environmental stimulus value. It can be seen that the stimulus value did not decrease directly but began to decline slowly after passing a certain value. This is because the *agent* did not respond to the order at the beginning, so the stimulus value of the sub-order kept increasing, and this finally led to the surge of the stimulus value of the overall environment. In addition, we can see that, although the environmental stimulus value will have an upward trend, it will decrease after rising to a certain extent and finally return to 0. When the environmental stimulus value returns to 0, it means that the total order has been fully allocated. Combined with Figure 4d, the remaining delivery orders, we can clearly see that every time the remaining delivery orders did not decrease, the environmental stimulation value rose accordingly, indicating that the order will directly lead to the environmental stimulation of the distribution of value changes. This is because the orders stimulate the value constantly under the influence of the increment of *δ*, thus increasing it. When the order stimulus value rises enough to attract the *agent* to respond to the order, the number of remaining orders to be delivered will also decrease. At the next moment, the order stimulus value to be allocated will return to 0 after the update, resulting in the reduction of the overall environmental stimulus value.

After 12 moments of such a cycle, both the environmental stimulus value and the number of remaining orders to be delivered returned to 0. At this point, the program ended, and, finally, 18 *agents* were assigned the distribution task of 18 sub-orders of the total order, indicating that the model algorithm program could finally complete the distribution task of the total order. In order to further verify the feasibility of the model, MATLAB was used to implement the model algorithm several times. It was found that, even though there were differences in the allocation process, the differences in the final results were not big, and the total orders could be fully allocated, which indicated that the model had good applicability and robustness.

Due to this chapter, the added cost of a carbon emissions constraint system has yet to be implemented at home. I conduct the simulation experiment under the assumption of a carbon tax rate of CNY 40/ton. So, in order to further explore the carbon tax policy-constrained internet freight order scheme of a distribution platform, the premise is to not affect economic benefits and to minimize the carbon emissions generated during operations. This paper intends to explore the relationship between the carbon tax rate and order distribution by comparing the differences in the model measurement indexes in the distribution results when carbon taxes with different tax rates are levied so as to find a more appropriate carbon tax rate range under the background of this problem. The specific difference results are shown in Table 6, and the change trend of relevant measurement indexes is shown in Figure 5 and Figure 6.

As can be seen from Table 6, when the carbon tax rate range is within [0, 150], there is little difference between the final total revenue in the whole order distribution process and the time when the final order distribution is completed. Only when the carbon tax rate rises to a certain level will there be a significant impact on the two. However, for the square value of no-load distance and carbon emissions, their changes show a trend of first increasing, then decreasing, and then increasing. Only the carbon emission cost increases monotonically with the increase in the carbon tax rate. This indicates that blindly increasing the carbon tax rate may not minimize the carbon emissions, so different industries may have different standards for a carbon tax.

This paper focuses on the order distribution of an online freight platform under the constraints of carbon tax policy, so it focuses on the study of carbon emissions. For carbon emissions, when there is no carbon cost constraint on freight transportation, that is, when w is 0, the carbon emissions will be 2004.3 kg, and the final total revenue will be about CNY 14,514. According to the data comparison in the table, when w is 40, the carbon emission reaches 1897.1, the lowest value of carbon emissions in Table 3, Table 4, Table 5 and Table 6, and the total income is about 14,376. It can be seen that the total income gap between the two cases is not large, but there is a significant difference in carbon emissions, which indicates that the model in this chapter can effectively reduce the carbon emissions generated by the order distribution process of the online freight platform.

As can be seen from Figure 5, with the increase in the carbon tax rate, the relationship between the square value of the empty load distance and carbon emissions is not purely linear. Under the common premise of increasing the carbon tax rate, looking at the change trend of carbon emissions in combination with the no-load distance, it can be found that when the no-load distance is the least, the carbon emissions are not the least, and the carbon emissions are not affected by the no-load distance. On the contrary, when the carbon tax rate was high, the no-load distance became larger, and the carbon emissions even decreased, so it can be shown that the no-load distance is not the main impact of carbon emissions in the order transportation problem. Variable, the greater impact is the carbon emissions generated during the transportation of different models. Therefore, it is a particularly important part to build a low-carbon and green network freight industry by arranging models that meet the needs of the orders to the greatest extent possible for orders with differentiated needs. In addition, when the carbon tax rate reaches a certain value, carbon emissions tend to be stable, and the cost of carbon emissions will be completely affected by the carbon tax rate. Combined with Figure 6, it can be seen that, with the increasing carbon tax rate, the cost of carbon emissions becomes higher and higher. In terms of revenue, the main part that decreases is the part of the carbon emissions cost that increases.

Therefore, on the premises that the total orders can be fully distributed and the change in revenue is small, it is the optimal carbon tax formulation scheme to choose the carbon tax rate with the lowest carbon emissions as the carbon tax rate under the background of this problem. Finally, observations were made through the above data, and MATLAB was used to fit the data of carbon tax rates and carbon emissions. Finally, the fitting function image of Figure 7 was obtained, and it was determined that the carbon tax rate interval most suitable for the research question in this paper is between 40 and 50.

In summary, a carbon tax is not purely negatively correlated with carbon emissions; there is an interval where carbon emissions reach a minimum value. The carbon tax and the no-load distance are not purely negatively correlated. When it increases to a certain limit, the carbon tax is positively correlated with the no-load distance; it can be clearly seen that the carbon tax has a significant positive correlation with the carbon cost of the online freight platform, and it is negatively correlated with its benefits. There is no exact impact relationship with carbon emissions, so it can be concluded that reasonable freight vehicle scheduling is more effective in reducing carbon emissions than purely reducing freight empty distances; in addition, there is no exact relationship between empty distances, carbon costs, and platform benefits. In the same way, it can be concluded that reasonable freight vehicle scheduling is more effective in reducing freight costs and generating income on the platform than purely reducing the empty freight distance. Finally, it can be intuitively concluded that carbon cost and platform revenue are significantly negatively correlated.

### 6.4. Comparative Analysis of Model Effects

In order to further verify the validity of the extended model and algorithm, the 0-1 integer programming model is used to solve the problem, and the results obtained by the extended model algorithm are compared and analyzed.

Because this chapter studies the carbon tax policy under the restriction of a freight network platform order allocation problem, lower carbon emissions and freight costs are essential to complete the order of distribution. Table 6 shows that the order allocation problem of the total benefit is small. In other words, the objective function can be interpreted as minimizing fuel costs and the cost of carbon emissions. The constructed model is as follows:(20)$$minZ=\sum X_{ij}\cdot \left(C_{ij}+E_{ij}\cdot w/1000\right)$$
(21)$$s.t.\;\;\;\sum_{j}X_{ij}=1\;\;\;\;$$
(22)$$\sum X_{ij}=18$$
(23)$$X_{ij}=1\;or\;0$$
where Equation (20) is the objective function. The freight cost considered in this chapter is mainly composed of the fuel cost and carbon emission cost, and $X_{ij}$ is the decision variable, which represents whether *agenti* will transport order J. At that time, $X_{ij}=1,$
*agenti*, and order J form a transport pairing relationship, so the fuel cost and carbon emission cost will be generated. Equation (21) indicates that an order can only be transported by one *agent*; Equation (22) indicates that the number of *agents* involved in the order transportation should be consistent with the order number, that is, the two meet the one-to-one matching relationship.

According to the order allocation result obtained by the 0-1 integer programming model, *Z* = 5582.9 and the total carbon emission $O_{1}$ = 1991.9, which is compared with the values (*Z* = 5892.6 and $O_{1}$ = 1897.2) obtained by the extended model in this chapter. It can be seen that the *Z* calculated by the 0-1 integer programming model is significantly smaller than that calculated by the extended order allocation model. However, the total carbon emission $O_{1}$ is larger than that calculated by the extended order allocation model. In terms of cost, the 0-1 integer programming model is superior to the extended order allocation model, but in terms of reducing carbon emissions, the extended order allocation model designed in this chapter is more advantageous.

In order to make the above conclusions more convincing, a comparative analysis is made regarding the carbon emissions and the total revenue of the final scheme under different carbon tax rates *w*. The specific results are shown in Table 7:

According to Table 7, showing the order distribution under the different carbon tax rates, when the carbon tax rate is low, the 0-1 integer programming of the optimal order allocation scheme, when the carbon tax rate rises to a certain extent, will affect the 0-1 integer programming model for allocation, but the final total revenue will also be discounted; this will greatly reduce the profitability of the final industry. In contrast, the extended order allocation model in this chapter can achieve the optimal solution level of 0-1 integer programming when the carbon tax rate is appropriate, that is, the goal of reducing carbon emissions can be achieved without greatly affecting the final profit of the industry. This further proves that the extended order allocation model constructed in this chapter and the algorithm program written have good effectiveness in solving the order allocation problem of network freight platforms under the constraints of carbon tax policy.

### 6.5. Parametric Analysis

By adjusting some parameters, the applicability of the model to this problem is further verified. Assume that the number of distribution outlets is adjusted to 10, and the number of car-hailing vehicles on the freight network is adjusted to 2 for each of the 6 models, that is, a total of 12 vehicles. The location of the car-hailing network of the freight network is within 5 km of factory X, the location of the distribution network is within 10–100 km of factory X, the order demand of the distribution network is within the range of 5 tons to 10 tons, and the order of the distribution network is within the range of 5 tons to 10 tons. The degree of urgency is in the range of 1 to 3. $D_{i}(i=1,\;2\ldots 12)=uniform\left(0,\;5\right),d_{j}\left(j=1,\;2\ldots 10\right)=uniform\left(10,\;100\right),G_{j}\left(j=1,\;2\ldots 10\right)=uniform\left(5,\;10\right),\varphi_{j}(j=1,\;2\ldots 10)=uniform\left(1,\;3\right)$. The rest of the parameters remain unchanged, and the running results are shown in Figure 8.

From Figure 8, the entire order allocation is finally completed at time t17. The overall scale of the order is smaller than the above scale, but the time consumption of the order allocation is not shorter than the above time consumption, which shows that the complexity of the order allocation problem is not purely proportional to the scale of the problem. Most of it is due to the complex balance of supply and demand and the adaptability of both supply and demand.

Since the reduction in carbon emissions is carried out on the basis of solving the problem of order allocation, in addition to the research on whether the model can effectively reduce carbon emissions, it is also necessary to explore whether the model can achieve the goal of order allocation. In order to further verify the effectiveness of the model for such problems, some parameters are adjusted again for the larger-scale order allocation problem. The number of distribution network points is now adjusted to 40, and the number of car-hailing vehicles on the freight network is adjusted to 12 for each of the 6 models, that is, the total number of car-hailing vehicles on the freight network is 72. The location of outlets and the order requirements are reset randomly. The changes of each indicator in the order allocation are shown in Figure 9:

As can be seen from Figure 9, the entire allocation process is still the same as the order allocation process of other scales. The amount of orders to be allocated decreases as the number of *agents* continues to respond to orders, and the busyness of group members is also increasing. The remaining order quantities all return to 0 at t18, and the order allocation is completed at this time, which shows that the model algorithm can also effectively solve the large-scale order allocation. By repeatedly verifying the applicability of the model algorithm in this chapter to the problem of the order allocation of different scales, the final experimental results show that the model and algorithm designed in this chapter have good effectiveness.

Through the comparison and analysis of Figure 4, Figure 8 and Figure 9, it is found that when orders of different sizes are allocated, the final running result of the model indicates that all orders can be completely allocated. The trend is ultimately reflected in the general direction of reducing the order volume in the overall order volume, which is in line with the real situation of real cargo transportation problems.

## 7. Conclusions

At present, the freight industry is gradually transforming into a digital information industry, and it is also facing the pressure of high carbon emissions. The research on the matching of vehicle and freight resources with carbon constraints is of great practical significance for the energy conservation and emission reduction of the freight industry. In order to promote the research of low-carbon freight transportation, this paper introduces carbon emissions and a carbon tax to generate carbon costs in the freight process and uses this as a constraint to conduct an order allocation optimization study on the network freight platform based on internet information technology. By extending the basic fixed threshold response model, the research defines the network freight platform carrier driver as an individual ant, classifies the models according to the difference in transportation costs, and sets a one-to-one corresponding response threshold for all ants and all sub-orders. The order stimulus value is set according to the remuneration given by each sub-order so as to use the stimulus value and the response threshold to dynamically coordinate and decide on the transportation arrangement. The specific implementation method of solving the problem in this paper is to code the designed model and algorithm program by MATLAB. Correspondingly, it is to change the model and algorithm of order matching on the online freight platform. This paper mainly models and simulates the order distribution of the online freight platform under the constraints of the carbon tax policy and compares it with the results obtained by using the 0-1 integer programming combined direct search method. The results show that the model has good flexibility, applicability, and robustness. In addition, by conducting several numerical simulation experiments after the establishment of different carbon tax rates, the comparative analysis of the differences in the model measurement indicators when different carbon tax rates are levied shows that a reasonable carbon tax can effectively reduce carbon emissions. In order to find the carbon tax rate range that satisfies the lowest carbon emissions under the condition that the total revenue does not change much, the functional relationship between the carbon tax rate and the carbon emissions is fitted, and the functional relationship between the carbon emissions and the carbon tax rate is obtained. Observing the image, it can be seen that the carbon tax rate range suitable for the background of this problem is within CNY 40–50/ton.

The results of the research content of this paper are in line with the expectations of the research purpose, but there are still shortcomings in the research process. In order to solve the problem of the order allocation of online freight platforms under the constraints of carbon tax policy, and in order to focus on the main influencing factors of related issues in this paper, some other secondary influencing factors are not considered. In addition, due to online freight originating in 2020, there are currently few works of literature on it. Some of the data used in this paper are not the latest data, and some of the data lack real data, so some data used in the calculation example are simulation data, that is, the calculation results of the example have certain limitations, and there may be deviations from the actual problem. In view of the above limitations, further detailed research can be carried out when the data of the online freight platform are more comprehensive. The model and solution algorithm designed in this paper have been verified by multiple numerical experiments, and the results show that it provides optimization support for the order allocation of online freight platforms under the background of carbon constraints and can provide some reference for the formulation of relevant emission reduction policies.

## Figures and Tables

**Figure 1 ijerph-19-10993-f001:**
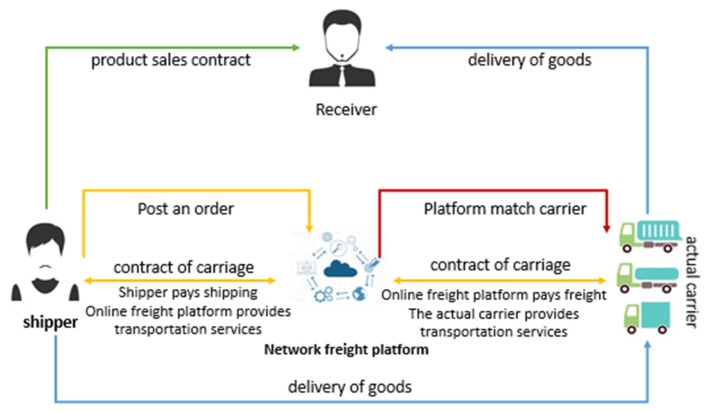
Operation principle diagram of the network freight platform.

**Figure 2 ijerph-19-10993-f002:**
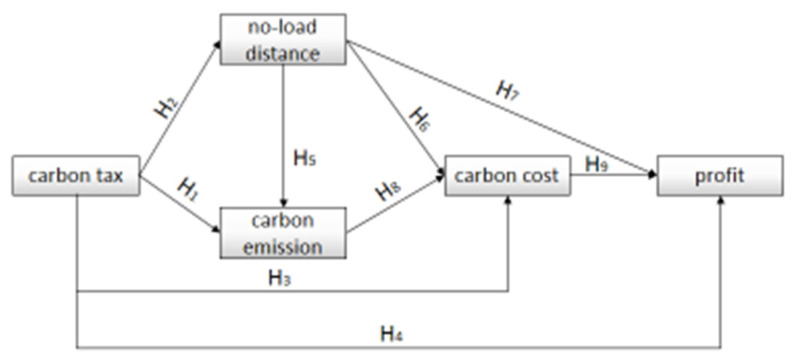
Relationship diagram between elements.

**Figure 3 ijerph-19-10993-f003:**
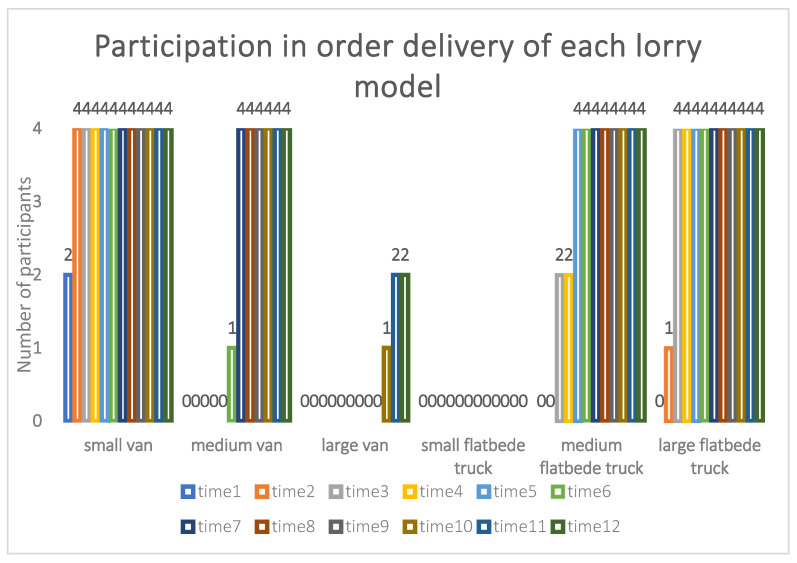
Order distribution of different models.

**Figure 4 ijerph-19-10993-f004:**
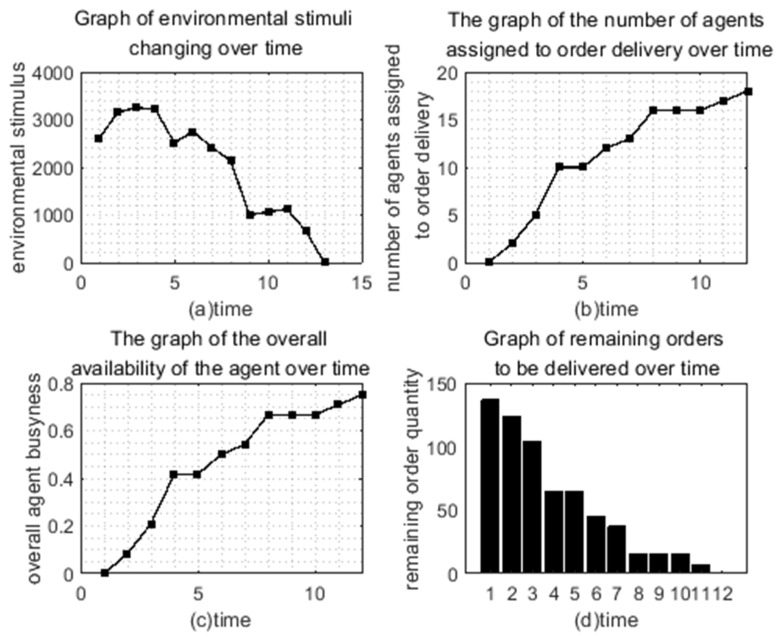
Order distribution macro picture.

**Figure 5 ijerph-19-10993-f005:**
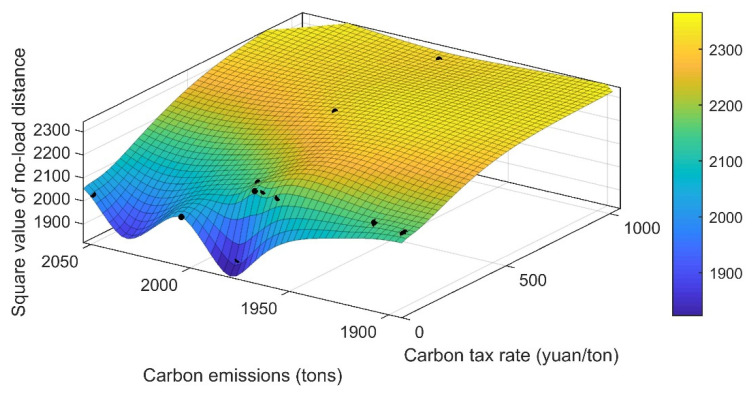
Carbon emissions and changes in square values of no-load distance under the influence of a carbon tax price.

**Figure 6 ijerph-19-10993-f006:**
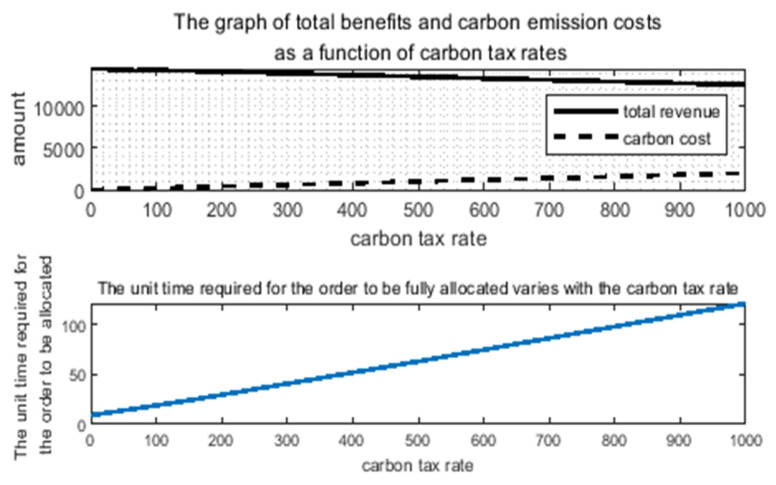
Changes in total benefits and carbon emission costs and order allocation time under the influence of carbon tax rates.

**Figure 7 ijerph-19-10993-f007:**
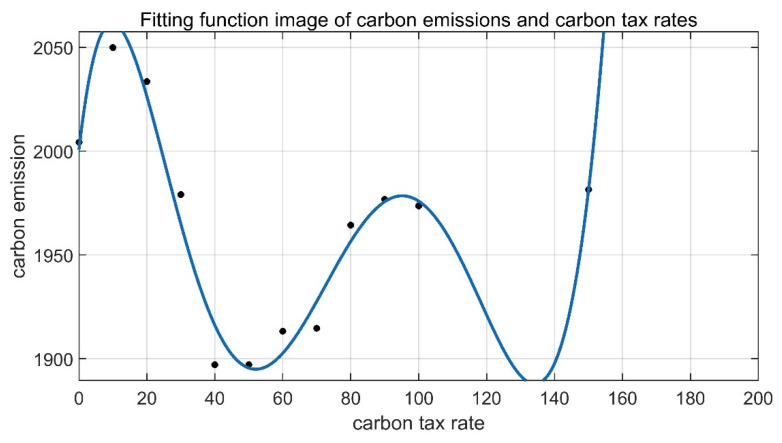
Fitting function of carbon emissions and the carbon tax rate.

**Figure 8 ijerph-19-10993-f008:**
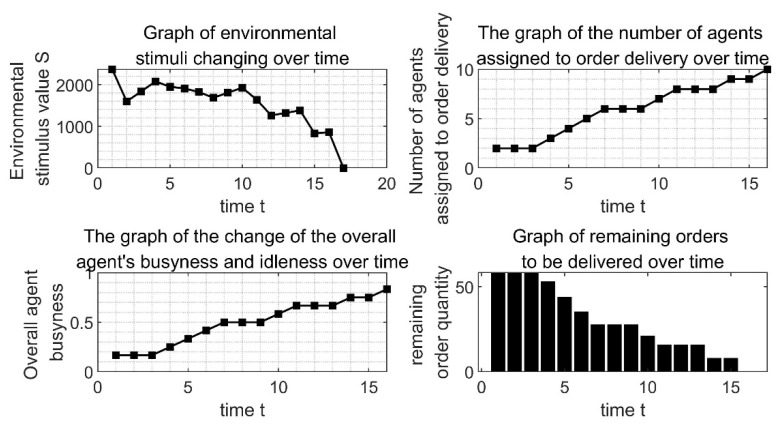
Change trend of key indicators during the allocation process.

**Figure 9 ijerph-19-10993-f009:**
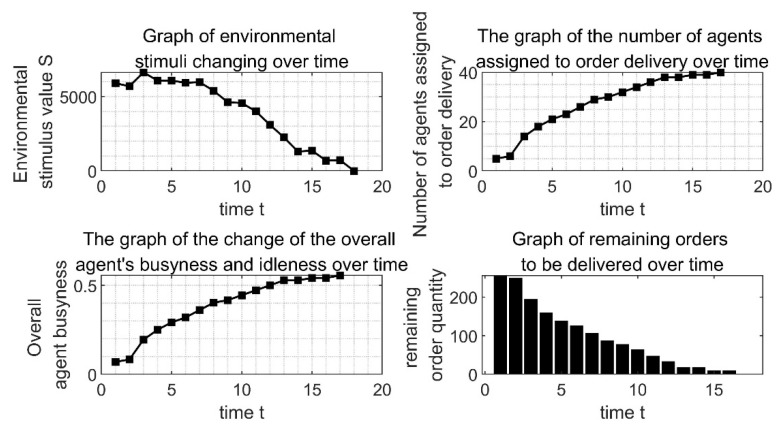
Changes in larger-scale order allocation indicators.

**Table 2 ijerph-19-10993-t002:** Freight booking operation pricing table.

Models	Load (kg)	Starting Price (5 km)	Continuation Price (Yuan/km)	Dead Weight (t)
small van	600	30	3	1
medium van	1000	50	4	1.5
large van	1500	60	4	2
small flatbed truck	1000	60	4	1
medium flatbed truck	1500	80	5	1.8
large flatbed truck	1800	100	5	2.2

Note: The data come from the delivery prices of some models of Huo Lala in 2021.

**Table 3 ijerph-19-10993-t003:** Relevant data of factories and distribution outlets.

	Distance between Factory and Distribution Network (km) $d_{j}$	Order Urgency $\varphi_{j}$	Order Volume $G_{j}$
Distribution network $T_{1}$	39.4	1.01	7.3
Distribution network $T_{2}$	35.2	1.08	8.1
Distribution network $T_{3}$	43.2	1.09	9.5
Distribution network $T_{4}$	23.6	1.18	5.8
Distribution network $T_{5}$	60.7	1.44	8.5
Distribution network $T_{6}$	57.4	1.45	7.8
Distribution network $T_{7}$	89.4	1.48	7.1
Distribution network $T_{8}$	32.2	1.71	9.4
Distribution network $T_{9}$	55.4	1.84	6.9
Distribution network $T_{10}$	24.2	1.87	5.8
Distribution network $T_{11}$	17.7	1.97	8.1
Distribution network $T_{12}$	58.0	2.07	8.1
Distribution network $T_{13}$	48.1	2.18	6.6
Distribution network $T_{14}$	16.0	2.45	9.0
Distribution network $T_{15}$	22.3	2.56	10.0
Distribution network $T_{16}$	9.6	2.68	5.6
Distribution network $T_{17}$	12.4	2.71	6.1
Distribution network $T_{18}$	13.6	2.92	7.1

Note: The data in the table order the serial number of distribution outlets according to the degree of emergency.

**Table 4 ijerph-19-10993-t004:** Distance between the factory and carrier driver.

	Distance between Factory and Freight Driver Receiving Unit $D_{i}$	Agent Unit Transportation Cost $c_{i}$	Carbon Emission Coefficient of Agent $\mu_{i}$	The Agent of Self-Respect $e_{i}$	Maximum Carrying Capacity of an Agent $v_{i}$
$agent_{1}$	1.09	0.5	0.151	0.6	1
$agent_{2}$	3.42	0.5	0.151	0.6	1
$agent_{3}$	4.03	0.5	0.151	0.6	1
$agent_{4}$	1.46	0.5	0.151	0.6	1
$agent_{5}$	1.37	0.5	0.151	0.6	1.5
$agent_{6}$	1.44	0.5	0.151	1	1.5
$agent_{7}$	2.62	0.5	0.151	1	1.5
$agent_{8}$	1.49	0.5	0.151	1	1.5
$agent_{9}$	4.81	0.5	0.151	1	2
$agent_{10}$	4.17	0.5	0.151	1	2
$agent_{11}$	4.53	0.5	0.151	1.5	2
$agent_{12}$	3.19	0.5	0.151	1.5	2
$agent_{13}$	3.81	0.4	0.155	1	1
$agent_{14}$	2.47	0.4	0.155	1	1
$agent_{15}$	1.83	0.4	0.155	1	1
$agent_{16}$	2.76	0.4	0.155	1	1
$agent_{17}$	4.82	0.4	0.155	1.5	1.8
$agent_{18}$	1.49	0.4	0.155	1.5	1.8
$agent_{19}$	2.88	0.4	0.155	1.5	1.8
$agent_{20}$	1.79	0.4	0.155	1.5	1.8
$agent_{21}$	2.57	0.4	0.155	1.8	2.2
$agent_{22}$	3.67	0.4	0.155	1.8	2.2
$agent_{23}$	2.88	0.4	0.155	1.8	2.2
$agent_{24}$	2.08	0.4	0.155	1.8	2.2

**Table 5 ijerph-19-10993-t005:** Operation results of the extended ant colony division of labor model.

Time *t*	Number of Minivans Already Engaged in Transport	Number of Medium Vans Already Engaged in Transport	Number of Large Vans Already Engaged in Transport	Number of Small Flatbed Vehicles Already Engaged in Transport	Number of Medium Flatbed Vehicles Already Engaged in Transport	Number of Large Flatbed Vehicles Already Engaged in Transport	Total Number of Agents Engaged in Transport	Quantity of Order to be Delivered	The Degree of the Driver’s Busy Time	Environmental Stimulus Value
*t0*	0	0	0	0	0	0	0	136.8	0	2615.5
*t*1	0	0	0	0	0	0	0	136.8	0	3155.5
*t*2	2	0	0	0	0	0	2	123.7	0.08	3254.8
*t*3	4	0	0	0	0	1	5	103.9	0.20	3215.3
*t*4	4	0	0	0	2	4	10	64.4	0.41	2512.3
*t*5	4	0	0	0	2	4	10	64.4	0.41	2752.3
*t*6	4	0	0	0	4	4	12	45.0	0.50	2421.0
*t*7	4	1	0	0	4	4	13	37.2	0.54	2144.0
*t*8	4	4	0	0	4	4	16	15.2	0.66	1004.7
*t*9	4	4	0	0	4	4	16	15.2	0.67	1064.7
*t*10	4	4	0	0	4	4	16	15.2	0.67	1124.7
*t*11	4	4	1	0	4	4	17	7.1	0.71	673.1
*t*12	4	4	2	0	4	4	18	0	0.75	0
*t*13	/	/	/	/	/	/	/	/	/	/

**Table 6 ijerph-19-10993-t006:** Differences of various indicators under the influence of the carbon tax rate.

	Measure	Carbon Emissions	Square of No-Load Distance	Carbon Cost	Final Gross Revenue	The Allocation Time is Finally Completed
A Carbon Tax Rate						
0		2004.3	2031.3	0	14,514.4	9
10		2049.9	2031.3	20.4	14,476.6	10
20		2033.5	1883.5	40.6	14,509.4	11
30		1979.1	1883.5	59.3	14,458.7	12
40		1897.1	2158.5	75.8	14,376.6	13
50		1897.1	2158.5	94.8	14,357.6	14
60		1913.2	2158.5	114.7	14,347.2	15
70		1914.6	2158.5	134.0	14,324.2	16
80		1964.3	2158.5	157.1	14,387.9	17
90		1976.8	2158.5	177.9	14,391.7	18
100		1973.6	2158.5	197.3	14,365.7	19
150		1981.5	2158.5	297.2	14,281.6	24
500		1979.5	2313.6	989.7	13,585.6	63
1000		1979.5	2313.6	1979.5	12,595.8	121

**Table 7 ijerph-19-10993-t007:** Comparison between the extended order allocation model and the 0-1 integer programming model.

	Measure	Extended Order Allocation Model’s Carbon Footprint	0-1 Integer Programming Model’s Carbon Emissions	Extended Order Allocation Model’s Final Total Revenue	0-1 Integer Programming Model’s Final Total Revenue
A Carbon Tax Rate					
0		2004.3	1991.9	14,514.4	14,766.0
10		2049.9	1991.9	14,476.6	14,746.1
20		2033.5	1991.9	14,509.4	14,726.2
40		1897.1	1991.9	14,376.6	14,686.4
100		1973.6	1991.9	14,365.7	14,566.8
1000		1979.5	1897.2	12,595.8	12,818.1

## Data Availability

Please refer to the suggested data or the MATLAB computer code at https://pan.baidu.com/s/1-EKQ1nuv699jSyFnXJEQrg (accessed on 9 August 2022).

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
