# Peer review of "The Order Allocation Problem and the Algorithm of Network Freight Platform under the Constraint of Carbon Tax Policy"

_ijerph, 2022, doi:10.3390/ijerph191710993_

Round 1

Reviewer 1 Report (Previous Reviewer 1)

Thank to Authors for improving the paper. I can see a lot of effort put into dodifying it.

However, not all of my comments were taken into account, so I still suggest the following changes:

- The literature review basically does not exist. A few of the references provided are not exhaustive. It definitely needs supplementing. Added three new items does not change a lot... 25 references it is not enought. It should be doubled or even more.

- thank you for adding research hypothesis. Have you been able to confirm them? All?

I still think, that the topic of the paper only partially refers to the subject of the journal and the special issue. Please note, that the main issues in the special issue are: carbon economy, economic development, carbon emission reduction behavior. In my opinion, this paper would be more appropriate for another journal, like Energies for example.

- There are no economic theories (and the special issue is relevant with economics).

- I still don't fully understand why such research methods were chosen? Why, in the opinion of the authors, they were the most appropriate for the study?

Why was this data and not others used in the study? Where are they coming from?

Author Response

Thank to Authors for improving the paper. I can see a lot of effort put into dodifying it.

However, not all of my comments were taken into account, so I still suggest the following changes:

- The literature review basically does not exist. A few of the references provided are not exhaustive. It definitely needs supplementing. Added three new items does not change a lot... 25 references it is not enought. It should be doubled or even more.

-On the basis of the previous one, the literature review was supplemented, and 25 papers were added for citations. The citations of the new papers covered the four fields of economic development[1,2], carbon economy[3,4,5,6,7,8,9,10,11,12,13,14], carbon constraints[15,16,17,18,19,20,21], cargo freight[22,23],and swarm intelligence labor division[24,25].

- thank you for adding research hypothesis. Have you been able to confirm them? All?

-The research hypotheses are summarized at the end of Section 6.3 as follows:

To sum up, carbon tax is not purely negatively correlated with carbon emissions, there is an interval where carbon emissions reach a minimum value; carbon tax and no-load distance are not purely negatively correlated. When it increases to a certain limit, the carbon tax is positively correlated with the no-load distance; it can be clearly concluded that the carbon tax has a significant positive correlation with the carbon cost of the online freight platform, and is negatively correlated with its benefits; no-load distance There is no exact impact relationship with carbon emissions, so it can be concluded that reasonable freight vehicle scheduling is more effective in reducing carbon emissions than purely reducing freight empty distances; in addition, there is no exact relationship between empty distances and carbon costs and platform benefits. In the same way, it can be concluded that reasonable freight vehicle scheduling is more effective in reducing freight costs and generating income on the platform than purely reducing the empty freight distance. Finally, it can be intuitively concluded that carbon cost and platform revenue are significantly negatively correlated.

I still think, that the topic of the paper only partially refers to the subject of the journal and the special issue. Please note, that the main issues in the special issue are: carbon economy, economic development, carbon emission reduction behavior. In my opinion, this paper would be more appropriate for another journal, like Energies for example.

- There are no economic theories (and the special issue is relevant with economics).

-In this revision, some supplements in economic theory have been added. I has supplemented the relevant theories and empirical studies on the relationship between economic development and industry development, the research status of carbon economy, and carbon cost accounting in the literature review.

- I still don't fully understand why such research methods were chosen? Why, in the opinion of the authors, they were the most appropriate for the study?

-Because the swarm intelligence division of labor is an overall intelligent behavior that a group of simple agents spontaneously cooperate to complete a common task. An optimal or approximate optimal solution to a problem-solving method.

In more detail, the division of labor in the ant colony can make the agent collectively complete all the division and cooperation from bottom to top without knowing the global demand. Matching degree, and then achieve a reasonable balance of division of labor. The division of labor is actually task allocation to a certain extent, and setting the response threshold and stimulus value for the corresponding subject can improve the flexibility of task allocation and complete task allocation under the condition that the overall demand is met to the greatest extent. The order allocation problem explored in this paper is essentially the matching of vehicle and cargo resources, which finally completes the order allocation by comprehensively processing the needs and expectations between the shipper and the carrier. Although the field of swarm intelligence labor division has been applied to many scenarios, the research on using swarm intelligence method to solve order allocation is still relatively scarce. And combined with the above applicability analysis, it can be seen that the order allocation problem has a high similarity with the ant colony labor division, so this paper chooses the threshold response model based on the ant colony labor division to analyze the order allocation problem of the online freight platform under the constraint of carbon tax. optimization.

-Why was this data and not others used in the study? Where are they coming from?

-After empirical investigation, after an example analysis of the data range in the problem, in order to ensure that the results of the problem solving are not random, the data taken in this paper are subject to different uniform distributions to more effectively verify the effectiveness of the model and algorithm, details as follows:.

Add citations

 [1] Dong K, Hochman G, Timilsina G R. Do drivers of CO2 emission growth alter overtime and by the stage of eco-nomic development?[J]. Energy Policy, 2020, 140: 111420.

[2]  Wang C, Kim Y-S, Kim C Y. Causality between logistics infrastructure and economic development in China[J]. Transport Policy, 2021, 100: 49–58.

[3]  Urry J. A low carbon economy and society[J]. Philosophical Transactions of the Royal Society A: Mathematical, Physical and Engineering Sciences, 2013, 371(1986): 20110566.

[4]  Xin X, Yuding W, Jianzhong W. The Problems and Strategies of the Low Carbon Economy Development[J]. Energy Procedia, 2011, 5: 1831–1836.

[5]  Zhang X-P, Cheng X-M. Energy consumption, carbon emissions, and economic growth in China[J]. Ecological Economics, 2009, 68(10): 2706–2712.

[6]  Heil M T, Selden T M. Carbon emissions and economic development: future trajectories based on historical experience[J]. Environment and Development Economics, 2001, 6(1): 63–83.

[7]  Boyce J K. Carbon Pricing: Effectiveness and Equity[J]. Ecological Economics, 2018, 150: 52–61.

[8]  Ionescu L. Transitioning to a Low-Carbon Economy[J]. Geopolitics, History, and International Relations, 2021, 13(1): 86-96.

[9]  Hailemariam A, Dzhumashev R, Shahbaz M. Carbon emissions, income inequality and economic development[J]. Empirical Economics, 2020, 59(3): 1139–1159.

[10] Yongping N. The Economic Thinking on Low Carbon Economy[J]. Energy Procedia, 2011, 5: 2368–2372.

[11] McEvoy D, Gibbs D C, Longhurst J W S. The employment implications of a low-carbon economy[J]. Sustainable Develop-ment, 2000, 8(1): 27–38.

[12] Sachs J, Stiglitz J, Mazzucato M, et al. Letter from economists: to rebuild our world, we must end the carbon economy[J]. The Guardian, 2020, 4.

[13] Jiang R, Zhou Y, Li R. Moving to a Low-Carbon Economy in China: Decoupling and Decomposition Analysis of Emission and Economy from a Sector Perspective[J]. Sustainability, 2018, 10(4): 978.

[14] Fu Yun, Ma Yonghuan, Liu Yijun, et al. Research on the development model of low-carbon economy [J]. China Population, Resources and Environment, 2008(03): 14–19.

[15] Elkins P, Baker T. Carbon Taxes and Carbon Emissions Trading[J]. Journal of Economic Surveys, 2002, 15(3): 325–376.

[16] Chen Z, Nie P. Effects of carbon tax on social welfare: A case study of China[J]. Applied Energy, 2016, 183: 1607–1615.

[17] Lu C, Tong Q, Liu X. The impacts of carbon tax and complementary policies on Chinese economy[J]. Energy Policy, 2010, 38(11): 7278–7285.

[18] Mandell S. Carbon emission values in cost benefit analyses[J]. Transport Policy, 2011, 18(6): 888–892.

[19] Ricke K, Drouet L, Caldeira K, et al. Country-level social cost of carbon[J]. Nature Climate Change, 2018, 8(10): 895–900.

[20] Liang Jin. Carbon Calculation and Carbon Management [J]. Science, 2022, 74(3): 1-4+69.

[21] Zhang C, Guo S, Tan L, et al. A carbon emission costing method based on carbon value flow analysis[J]. Journal of Cleaner Production, 2020, 252: 119808.

[22] Fernández-Portillo A, Almodóvar-González M, Coca-Pérez J L, et al. Is Sustainable Economic Development Possible Thanks to the Deployment of ICT?[J]. Sustainability, 2019, 11(22): 6307.

[23] Pan X-X, Chen M-L, Ying L-M, et al. An empirical study on energy utilization efficiency, economic development, and sustainable management[J]. Environmental Science and Pollution Research, 2020, 27(12): 12874–12881.

[24] Baker P M. The Division of Labor: Interdependence, Isolation, and Cohesion in Small Groups[J]. Small Group Behavior, 1981, 12(1): 93–106.

[25] Deng X, Li J, Liu E, et al. Task allocation algorithm and optimization model on edge collaboration[J]. Journal of Systems Architecture, 2020, 110: 101778.

Reviewer 2 Report (Previous Reviewer 2)

Yes, improvements have been made. Congratulations and good luck for the future!

Author Response

-Yes, improvements have been made. Congratulations and good luck for the future!

Response: thanks!

This manuscript is a resubmission of an earlier submission. The following is a list of the peer review reports and author responses from that submission.

Round 1

Reviewer 1 Report

In my opinion, the topic of the paper only partially refers to the subject of the journal and the special issue. Please note, that the main issues in the special issue are: carbon economy, economic development, carbon emission reduction behavior.

In my opinion, this paper would be more appropriate for another journal, like Energies for example.

The literature review basically does not exist. A few of the references provided are not exhaustive. It definitely needs supplementing.

There are no economic theories.

Basically there is no discussion part. There is only a presentation of the research results.

I don't fully understand why such research methods were chosen? Why, in the opinion of the authors, they were the most appropriate for the study?

Why was this data and not others used in the study? Where are they coming from?

I don't understand the conclusion: "Some other secondary factors are not taken into account in the model. In addition, some data used are not the latest published data, and some data lack real data. Some data used in the calculation of examples are simulation data, so there are certain limitations in the calculation of examples, and there may be deviations from actual problems." The Authors basically exlude their study. Do I understand correctly?

The authors did not indicate the research hypothesis, and the research goal for me is not very specific.

Footnotes and bibliography are correctly formulated.

The language of the article is mature, correct, adequate.

Reviewer 2 Report

I've carefully read your paper and it's extremely intersting, congratulations on your work.

I have a few minor suggestions:

-please improve the literature review on the topic, by including more recent papers from the main flow of top journals. You have 2 references from 2020 and 1 from 2022, ok, but please expand your included citations from the main research flow;

-you've used MatlabR2018, great! I've recognized its surface representations in Figure 6. I haven't used Matlab in at least 1o years by now, but I'm positive you can save its images at a better resolution. Please do.

-I'd suggest you moved Figure 2 and Figures 3 within an Appendix.

Great work, once again! It may be published as soon as you take care of the above mentioned aspects. Thank you!

Reviewer 3 Report

Congratulations, excellent paper. I just wonder if you can better explain how this theoretical model can be transformed in an operational one.

REgarding what you said on your conclusions (copy below), the model es oversimplified and it is not operative. Is that true? Can you explain on the article this constrains and how can be solved?

In this paper, the research content of expected results conform to the research pur- 679
pose, but still exist deficiencies in the process of study, because my ability is limited, the 680
freight of the carbon tax policy under the restriction of network platform orders allocation 681
model and the network driver in the profit distribution model of solving freight platform, 682
to simplify the model, only focus on the related issues in this paper the main influence 683
factors, Some other secondary factors are not taken into account in the model. In addition, 684
some data used are not the latest published data, and some data lack real data. Some data 685
used in the calculation of examples are simulation data, so there are certain limitations in 686
the calculation of examples, and there may be deviations from actual problems. In this 687
paper, the freight scheduling problem of short-distance urban transport and the profit 688
distribution problem after dispatching are considered, but long-distance and multimodal 689
transport are not considered. Further carbon reduction studies can be carried out for trans- 690
city transport and multimodal transport.